# WHY SACRIFICE MAJORITY NODES?: IMPROVING IMBALANCED NODE CLASSIFIERS VIA CLASS-BALANCED GRAPH GENERATION

## ABSTRACT

Class imbalance is prevalent in real-world data, often leading to a deterioration in a classifier's generalization performance, especially on minority classes. Since graph-structured data is no exception, many efforts have been made to tackle imbalanced node classification by focusing on minority classes, leading to improved overall performance in imbalanced node classification. However, we find that these methods boost minority recall at the expense of degrading majority recall, a trade-off that has been overlooked. To address this issue, we propose Class Balancing Graph Generation (CBGG), a novel framework that prevents imbalanced node classifiers from sacrificing prediction power on majority classes. CBGG trains classifiers on high-quality synthetic graphs with class-balanced nodes, thereby tightening their generalization bounds across all classes. Extensive experimental results demonstrate that CBGG not only overcomes the majority-sacrifice pitfall of prior work but also significantly outperforms state-of-the-art imbalanced node classification methods across seven benchmark datasets.

## 1 INTRODUCTION

Semi-supervised node classification on graph-structured data is a fundamental task with wide applications in domains such as e-commerce and bioinformatics (Kipf & Welling, 2017; Veličković et al., 2018; Hamilton et al., 2017). However, real-world graphs often exhibit severe class imbalance (Bojchevski & Günnemann, 2018; Sen et al., 2008; Shchur et al., 2018), which poses significant challenges for learning. For instance, in citation networks, certain research fields may contain only a few labeled nodes, making it difficult for models to generalize to those under-represented classes. This class imbalance degrades classification performance by biasing the model toward majority classes and increasing error rates on minority nodes.

To mitigate this issue, existing methods typically increase the relative presence of minority nodes—either by assigning higher weights to minority samples (Yuan & Ma, 2012) or by synthesizing additional minority nodes (Zhao et al., 2021; Park et al., 2021a; Li et al., 2023). However, we identify a common trade-off in prior methods: improving recall for minority classes often comes at the cost of reduced majority-class recall. As shown in Figure 1(a), these methods consistently increase recall for minority classes but simultaneously reduce recall for majority classes compared to a standard GNN classifier (Veličković et al., 2018), highlighting an inherent trade-off.

The trade-off we identify motivates us to enhance imbalanced node classifiers through *graph-level generation*, which enables training on synthetic graphs that closely resembles the input graph. We hypothesize that generating class-balanced synthetic nodes with high label-predictive consistency plays a key role in reducing the generalization gap of classifiers, which we validate through both theoretical and empirical analyses. To implement this idea in practice, we draw inspiration from recent advances in diffusion-based graph generation (Jo et al., 2022; Vignac et al., 2023; Chen et al., 2023). However, these methods are primarily designed for graph synthesis and are not tailored to imbalanced node classification tasks.

In this paper, we propose Class Balancing via Graph Generation (CBGG), a novel plug-in framework that addresses the trade-off between minority- and majority-class performance in imbalanced node classification. CBGG begins by training an initial node classifier (e.g., GraphENS (Park et al.,

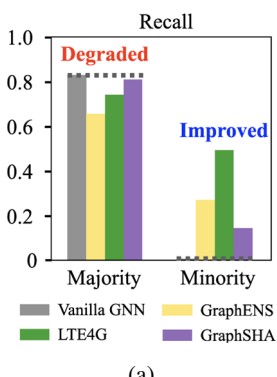
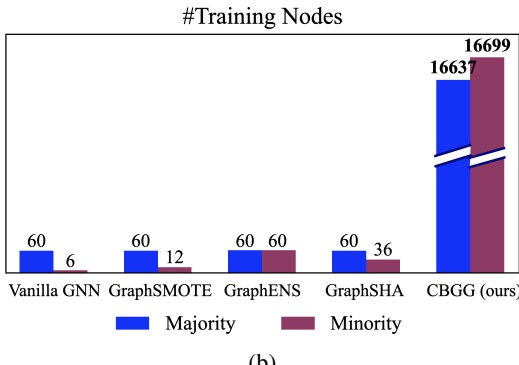

|     |     |
| --- | --- |
| (a) | (b) |

Figure 1: Results on CiteSeer dataset: (a) recall of existing methods for majority and minority nodes, and (b) the number of training nodes for majority and minority classes.

2021a) or GraphSHA (Li et al., 2023)). CBGG then trains a new diffusion-based graph generator that produces synthetic graphs with class-balanced nodes. The generator is optimized with a supervised contrastive classification loss and is conditioned on a soft label distribution obtained from the initial node classifier. Lastly, a final, improved node classifier is trained on both the original graph and multiple diverse class-balanced graphs produced by the generator. This approach enables the classifiers to train on a substantially larger number of majority and minority nodes, as shown in Figure 1(b), thereby boosting classification performance on minority nodes without sacrificing the performance of majority nodes. We provide a theoretical analysis showing that CBGG reduces the generalization gap and empirically demonstrate that CBGG effectively mitigates the majority–minority trade-off. Finally, CBGG significantly outperforms state-of-the-art imbalanced node classification methods across seven benchmark datasets.

In summary, our contributions are as follows: **(i)** We identify the minority–majority trade-off inherent in existing imbalalanced node classification methods, which has been underexplored. **(ii)** We introduce a new paradigm for imbalanced node classification based on graph-level generation. **(iii)** We theoretically highlight that CBGG mitigates the generalization gap by increasing the number of synthetic samples through graph-level generation and minimizing the synthetic noise rate through the supervised contrastive loss. **(iv)** Through extensive experiments on seven benchmark datasets, we demonstrate that CBGG achieves significant performance gains over state-of-the-art methods for imbalanced node classification.

## 2 RELATED WORK

**Class-Imbalance Learning.** Methods for class imbalance learning (Chen et al., 2024), including its special case, long-tailed learning (Zhang et al., 2025), can be broadly categorized into two approaches: algorithm-level and data-level. The algorithm-level approach focuses on modifying the training process—such as introducing reweighting strategies (Huang et al., 2016; Wang et al., 2017; Cui et al., 2019; Cao et al., 2019; Ren et al., 2020; Hong et al., 2021) or adjusting loss function (Lin et al., 2017; Park et al., 2021b)—to make the model more sensitive to minority classes, without altering the original data distribution. In contrast, the data-level approach addresses class imbalance by modifying the training data distribution itself—typically through repeatedly sampling minority samples (*i.e.* oversampling) (Chawla et al., 2002; Mullick et al., 2019; Kim et al., 2020; Park et al., 2022) or selectively removing a portion of majority samples (*i.e.* undersampling) (Mani & Zhang, 2003; Van Hulse et al., 2007; Kang et al., 2016)—to create a more balanced dataset for training.

**Imbalanced Node Classification.** Tailored techniques for imbalanced node classification on graphs (Ma et al., 2025; Liu et al., 2025) have been developed under the same algorithm-level and data-level categorization used in general class-imbalance learning. As the algorithm-level approach, ReVar (Yan et al., 2023) introduces variance-based regularization into the loss function, while LTE4G (Yun et al., 2022) trains multiple expert graph neural networks (GNNs) for different imbalanced node subsets and distills their knowledge into class-wise student models. As the data-level approach, most methods for imbalanced node classification (Park et al., 2021a; Zhao et al.,

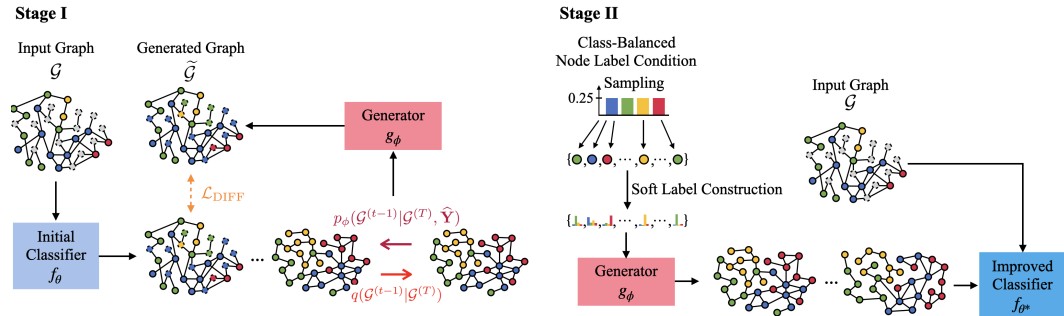

Figure 2: An overview of Class Balancing via Graph Generation (CBGG).

2021; Li et al., 2023; 2024b) have been based on oversampling, which synthesizes minority nodes to mitigate class imbalance. Although these methods improve performance on minority classes by generating synthetic minority nodes, we find that this comes at the cost of degrading performance on majority classes. Unlike oversampling-based methods, our graph generation-based approach effectively resolves this issue by generating a large number of both majority and minority nodes.

**Graph Generation.** With the development of deep generative models, many data-driven approaches for graph structure generation have been developed (Kipf & Welling, 2016; You et al., 2018b; Bojchevski et al., 2018; Li et al., 2018; Liao et al., 2019). Deep generative models have also been applied to molecular graph generation (Jin et al., 2018; Liu et al., 2018; You et al., 2018a; De Cao & Kipf, 2018), where each graph typically contains a few dozen nodes and a single categorical attribute per node. As the trend in deep generative models has shifted toward diffusion models (Ho et al., 2020; Rombach et al., 2022), this paradigm has also inspired the development of diffusion-based methods tailored for graph structure and molecule generation (Niu et al., 2020; Jo et al., 2022; Vignac et al., 2023; Chen et al., 2023; Kong et al., 2023; Ketata et al., 2025). Among them, Graph-Maker (Li et al., 2024a) improves scalability by replacing the graph transformer encoder Dwivedi & Bresson (2020) with a message-passing neural network (MPNN) Gilmer et al. (2017) and adopting a minibatch strategy. However, they are not designed for semi-supervised node classification.

## 3 PROPOSED METHOD

### 3.1 PROBLEM DEFINITION

The problem of node classification considers a graph $\mathcal{G} = (\mathcal{V}, \mathcal{E}, \mathbf{X})$, where $\mathcal{V} = \{v_1, \ldots, v_N\}$ is the set of $N$ nodes, $\mathcal{E}$ is the set of edges, and $\mathbf{X} \in \mathbb{R}^{N \times F}$ is a node feature matrix, with $F$ being the number of feature channels. Specifically, $\mathbf{X}_{i,a}$ represents the $a$-th channel feature value of $v_i$. To represent the structure of $\mathcal{G}$, we use the adjacency matrix $\mathbf{A} \in \{0,1\}^{N \times N}$. $\widetilde{\mathbf{A}} \in \{0,1\}^{N \times N \times 2}$ is a one-hot encoded tensor of $\mathbf{A}$, where the last dimension represents the presence or absence of an edge. $\widetilde{\mathbf{X}} \in \mathbb{R}^{N \times F \times B_\mathbf{X}}$ is a one-hot encoding of $\mathbf{X} \in \mathbb{R}^{N \times F}$, where $B_\mathbf{X}$ denotes the possible classes of all categorical attributes. Since each node is also associated with a class label, $\mathbf{Y} \in \{0,1\}^{N \times C}$ denotes a node label matrix, where $C$ is the number of classes. The $i$-th row of $\mathbf{Y}$, denoted as $\mathbf{Y}_{i,:}$, is a one-hot vector indicating the ground-truth label of node $v_i$. All nodes in $\mathcal{V}$ are partitioned into $\mathcal{V}_{\text{train}}$, $\mathcal{V}_{\text{val}}$, and $\mathcal{V}_{\text{test}}$, which represent the training, validation, and test sets, respectively. The goal of node classification is to accurately predict the labels of nodes in $\mathcal{V}_{\text{test}}$ by training a model on labeled nodes in $\mathcal{V}_{\text{train}}$, using both $\mathbf{X}$ and $\mathbf{A}$. In this work, we focus on imbalanced node classification, where the number of labeled nodes significantly differs across classes. That is, we consider class imbalance within the training set $\mathcal{V}_{\text{train}}$, which contains the labeled nodes available during training.

### 3.2 OVERVIEW OF CBGG

We introduce CBGG, a two-stage plug-in framework designed to mitigate class imbalance in semi-supervised node classification. As illustrated in Figure 2, CBGG consists of two stages. In the first stage, CBGG trains a new diffusion-based graph generator designed for node classification to

reconstruct the input graph, using the predictions of the initial classifier $f_\theta$. In the second stage, the classifier is retrained as $f_{\theta^*}$ using class-balanced synthetic graphs generated by the trained generator.

## 3.3 Stage I: Training a Graph Generation Model

Given a node classifier $f_\theta$ such as GraphSMOTE (Zhao et al., 2021), GraphENS (Park et al., 2021a), and GraphSHA (Li et al., 2023), our CBGG aims to reduce the generalization error of $f_\theta$ via graph-level generation. To accomplish this, we train a new label-conditioned diffusion model $g_\phi$ that learns to recover the input graph $\mathcal{G} = (\mathcal{V}, \mathcal{E}, \mathbf{X})$ via a denoising diffusion process.

We generate $\widehat{\mathbf{Y}}_v$ by replacing $\mathbf{Y}_v$ of unobserved nodes $v \in \mathcal{V}_{valid} \cup \mathcal{V}_{test}$ with the soft label predictions (i.e., probability distributions) of $f_\theta$. While labeled nodes retain their ground-truth one-hot labels, the rest are filled with these soft label probability vectors. The primary advantage of this approach is that it allows us to capture and incorporate the uncertainty inherent in the classifier's predictions into the graph generation process. This ultimately helps to reduce the generalization error of the node classifier $f_\theta$. Our diffusion model $g_\phi$ consists of a **forward process** and a **reverse process**.

**Forward Process.** In the forward process, the given graph $\mathcal{G}$ is progressively corrupted by perturbing its attributes $\mathbf{X}$ and edges $\mathbf{A}$ over multiple steps. Subsequently, during the reverse process, $g_\phi$ is trained to perform one-step denoising. We follow the asynchronous strategy designed by Graph-Maker (Li et al., 2024b)). We define $\mathcal{T}_{\widetilde{\mathbf{X}}} = \{t_{\widetilde{\mathbf{X}}}^1, \ldots, t_{\widetilde{\mathbf{X}}}^{T_{\widetilde{\mathbf{X}}}}\} \subseteq [T]$ as the set of time steps dedicated to corrupting node attributes, and similarly, $\mathcal{T}_{\widetilde{\mathbf{A}}} = \{t_{\widetilde{\mathbf{A}}}^1, \ldots, t_{\widetilde{\mathbf{A}}}^{T_{\widetilde{\mathbf{A}}}}\} \subseteq [T]$ for edge corruption.

During the forward process, we define the corruption transition $\overline{\mathbf{Q}}_{\widetilde{\mathbf{X}}_d}^{(t)} \in \mathbb{R}^{B_{\mathbf{X}} \times B_{\mathbf{X}}}$ and $\overline{\mathbf{Q}}_{\widetilde{\mathbf{A}}}^{(t)} \in \mathbb{R}^{2 \times 2}$ as

$$\overline{\mathbf{Q}}_{\widetilde{\mathbf{X}}_d}^{(t)} = \bar{\alpha}_{\gamma_{\widetilde{\mathbf{X}}}}(t) \cdot \mathbf{I}_{B_{\mathbf{X}} \times B_{\mathbf{X}}} + \left(1 - \bar{\alpha}_{\gamma_{\widetilde{\mathbf{X}}}}(t)\right) \cdot \mathbf{1}_{B_{\mathbf{X}}} \cdot \mathbf{m}_{\widetilde{\mathbf{X}}_d}^\top, \tag{1}$$

$$\overline{\mathbf{Q}}_{\widetilde{\mathbf{A}}}^{(t)} = \bar{\alpha}_{\gamma_{\widetilde{\mathbf{A}}}}(t) \cdot \mathbf{I}_{2 \times 2} + \left(1 - \bar{\alpha}_{\gamma_{\widetilde{\mathbf{A}}}}(t)\right) \cdot \mathbf{1}_2 \cdot \mathbf{m}_{\widetilde{\mathbf{A}}}^\top. \tag{2}$$

where $\bar{\alpha}_{\gamma_Z}(t) = \cos^2\left(\frac{\pi}{2} \cdot \frac{\gamma_Z(t)}{|\mathcal{T}_Z| + s}\right)$ is a time-dependent noise coefficient, and $Z \in \{\widetilde{\mathbf{X}}, \widetilde{\mathbf{A}}\}$. $\mathbf{m}_{\widetilde{\mathbf{X}}_d}$ denotes a $B_{\mathbf{X}}$-dimensional column vector that represents the empirical marginal distribution of the $d$-th node attribute, and $\mathbf{m}_{\widetilde{\mathbf{A}}}$ a two-dimensional column vector that represents the empirical marginal distribution of edge existence. $\mathbf{I}_{m \times m} \in \{0, 1\}^{m \times m}$ is the identity matrix, $\mathbf{1}_m$ is an $m$-dimensional column vector of ones, and $\top$ denotes transpose. To ensure that the corrupted adjacency tensor $\widetilde{\mathbf{A}}^{(t)}$ remains symmetric and reflects an undirected graph, the transition matrix $\overline{\mathbf{Q}}_{\widetilde{\mathbf{A}}}^{(t)}$ is applied only to the upper triangular part of $\widetilde{\mathbf{A}}$, and the full matrix is reconstructed via symmetrization after sampling.

Using these matrices, the forward process corrupts the graph as:

$$q(\widetilde{\mathbf{X}}_{i,d}^{(t)} \mid \widetilde{\mathbf{X}}_{i,d}) = \widetilde{\mathbf{X}}_{i,d} \cdot \overline{\mathbf{Q}}_{\widetilde{\mathbf{X}}_d}^{(t)} \quad \text{for } i \in [N], \ d \in [F], \tag{3}$$

$$q(\widetilde{\mathbf{A}}^{(t)} \mid \widetilde{\mathbf{A}}) = \widetilde{\mathbf{A}} \cdot \overline{\mathbf{Q}}_{\widetilde{\mathbf{A}}}^{(t)}, \tag{4}$$

This process gradually transforms the clean node features and adjacency matrix into their noisy counterparts across time steps.

**Reverse Process.** In the reverse process, $g_\phi$ denoises the graph one step at a time. To guide the generation toward class-consistent structure, we condition each node's input on its label embedding. This conditioning is implemented by concatenating soft-label label vectors. The resulting conditioned matrix is used as input to the denoising network $g_\phi$ at each timestep. Accordingly, the denoising distribution becomes conditioned on the label matrix $p_\phi(\mathcal{G}^{(t-1)} \mid \mathcal{G}^{(t)}, t, \widehat{\mathbf{Y}})$. Once trained, $g_\phi$ generates a graph by iteratively denoising a noisy sample drawn from an empirical prior distribution over node features and edges, formally defined as $\prod_{v=1}^N \prod_{d=1}^F \mathbf{m}_{\widetilde{\mathbf{X}}_d} \cdot \prod_{1 \leq u < v \leq N} \mathbf{m}_{\widetilde{\mathbf{A}}}$. This denoising process is performed step-by-step, where the joint distribution at each timestep is

factorized as follows:

$$p_\phi(\mathcal{G}^{(t-1)} \mid \mathcal{G}^{(t)}, \widehat{\mathbf{Y}}) = \prod_{i=1}^{N} \prod_{d=1}^{F} p_\phi(\widetilde{\mathbf{X}}_{i,d}^{(t-1)} \mid \mathcal{G}^{(t)}, \widehat{\mathbf{Y}}_i)$$
$$\cdot \prod_{1 \leq j < i \leq N} p_\phi(\widetilde{\mathbf{A}}_{j,i}^{(t-1)} \mid \mathcal{G}^{(t)}, \widehat{\mathbf{Y}}_i). \tag{5}$$

This procedure predicts clean features $\mathbf{X}$ and structure $\mathbf{A}$ from the noisy inputs and conditioning labels.

To compute each denoising term, we marginalize over the model's categorical predictions. Specifically, for each node attribute dimension, we compute:

$$p_\phi(\widetilde{\mathbf{X}}_{i,d}^{(t-1)} \mid \mathcal{G}^{(t)}, \widehat{\mathbf{Y}}_i) = \sum_{x_d} q(\widetilde{\mathbf{X}}_{i,d}^{(t-1)} \mid \widetilde{\mathbf{X}}_{i,d}^{(t)}) \hat{p}_{i,d}^X(x_d | \widehat{\mathbf{Y}}_i), \tag{6}$$

where $\hat{p}_{i,d}^X = \text{softmax}(W\mathbf{Z}_i)$ and $\mathbf{Z} = g_\phi(\mathcal{G}^{(t)}, t, \widehat{\mathbf{Y}})$ is a node representation matrix. $W$ is a trainable matrix and $g_\phi$ is a message-passing neural network.

To optimize parameters $\phi$, we minimize a hybrid loss that combines a reconstruction loss and a supervised contrastive loss. The full objective is defined as:

$$\mathcal{L}_{\text{DIFF}} = \alpha \mathcal{L}_{\text{SC}} + \sum_{1 \leq i \leq N} \text{CE}(\mathbf{X}_i, \widetilde{\mathbf{X}}_i) + \sum_{1 \leq i,j \leq N} \text{CE}(\mathbf{A}_{ij}, \widetilde{\mathbf{A}}_{ij}), \tag{7}$$

where $\alpha$ is a hyperparameter and CE is a cross-entropy loss.

The first term $\mathcal{L}_{\text{SC}}$ is a kind of supervised contrastive loss (Khosla et al., 2020) that encourages nodes with the same labels to be close:

$$\mathcal{L}_{\text{SC}} = \sum_{i \in \mathcal{V}_{train}} \frac{-1}{|P(i)|} \sum_{p \in P(i)} \log \frac{\exp(\mathbf{Z}_i \cdot \mathbf{Z}_p / \tau)}{\sum_{a \in \mathcal{V}_{train} \setminus \{i\}} \exp(\mathbf{Z}_i \cdot \mathbf{Z}_a / \tau)}, \tag{8}$$

where $P(i) = \{p \in \mathcal{V}_{train} \setminus \{i\} \mid \mathbf{Y}_p = \mathbf{Y}_i\}$, and $\tau$ is a temperature hyperparameter. Our $\mathcal{L}_{\text{SC}}$ uses labeled nodes as anchors.

### 3.4 STAGE II: ENHANCING NODE CLASSIFIER VIA CLASS-BALANCED SYNTHETIC GRAPHS

**Sampling Synthetic Graphs.** To enhance the classifier $f_\theta$, CBGG generates class-balanced synthetic graphs using the diffusion model $g_\psi$. This is achieved by repeatedly applying the reverse process of $g_\psi$. Unlike prior diffusion-based models that follow the empirical class distribution of the input graph, our approach conditions generation on a uniform class prior. This design increases the minimum number of training samples across classes ($n_{\min}$), thereby tightening the generalization bound between true and empirical risks, as discussed in Section 3.5.

To achieve this, we employ a per-class buffer $\{\mathcal{B}_c\}_{c=1}^{C}$ of soft label distributions produced by the pretrained teacher $f_\theta$. Each $\mathcal{B}_c$ stores probability vectors $p_{f_\theta}(\cdot \mid v) \in \Delta^{C-1}$ collected from nodes $v$ that are assigned to class $c$ (i.e., $\arg\max_y p_{f_\theta}(y \mid v) = c$).

Let $\{\widetilde{\mathcal{G}}_k\}_{k=1}^{K}$ be the set of synthetic graphs to be generated. For each $k$, we first construct a soft-label matrix $\mathbf{Y}_k \in \mathbb{R}^{N \times C}$ by independently sampling soft label vectors from the buffers as follows:

$$(\mathbf{Y}_k)_{i,:} \sim \text{Uniform}(\mathcal{B}_{c_i}), \quad c_i \sim \text{Uniform}(\{1, \ldots, C\}), \tag{9}$$

The sampled matrix $\mathbf{Y}_k$ serves as the conditioning signal that enforces class balance during generation while incorporating calibrated uncertainty.

We then sample an initial noisy graph $\widetilde{\mathcal{G}}_k^{(T)} = (\widetilde{\mathbf{X}}_k^{(T)}, \widetilde{\mathbf{A}}_k^{(T)})$ from a predefined prior distribution over features and adjacency structures. Starting from this noisy graph, the model performs reverse diffusion in $T$ steps:

$$\widetilde{\mathcal{G}}_k^{(t-1)} \sim p_\phi(\widetilde{\mathcal{G}}_k^{(t-1)} \mid \widetilde{\mathcal{G}}_k^{(t)}, t, \mathbf{Y}_k). \tag{10}$$

where $t = T, T-1, \ldots, 1$. At each step, the model predicts cleaner features and structure conditioned on the current noisy graph, the timestep, and the label matrix.

Table 1: Step-class imbalanced node classification results on three citation networks. **Bold** marks the best scores, while underlining denotes the second-best.

| Method | Cora (Imb. class num = 3) | | | Citeseer (Imb. class num = 3) | | | Pubmed (Imb. class num = 1) | | |
|---|---|---|---|---|---|---|---|---|---|
| **Imb. Ratio** ($\rho = 0.05$) | GMeans | bAcc | Macro-F1 | GMeans | bAcc | Macro-F1 | GMeans | bAcc | Macro-F1 |
| OverSampling | $70.4 \pm 2.9$ | $53.7 \pm 1.6$ | $47.0 \pm 2.2$ | $63.3 \pm 1.7$ | $45.0 \pm 2.2$ | $35.2 \pm 2.5$ | $66.7 \pm 5.4$ | $56.8 \pm 6.8$ | $54.9 \pm 7.3$ |
| Reweight | $76.7 \pm 2.2$ | $62.7 \pm 3.2$ | $59.5 \pm 5.4$ | $64.9 \pm 1.7$ | $47.2 \pm 2.3$ | $38.3 \pm 3.7$ | $70.5 \pm 0.6$ | $61.6 \pm 0.7$ | $54.1 \pm 2.8$ |
| SMOTE | $69.5 \pm 1.6$ | $52.5 \pm 2.1$ | $45.4 \pm 4.9$ | $62.8 \pm 2.1$ | $44.4 \pm 2.8$ | $34.2 \pm 3.6$ | $66.5 \pm 2.0$ | $56.5 \pm 2.5$ | $48.2 \pm 3.0$ |
| GraphSMOTE | $73.3 \pm 0.6$ | $57.8 \pm 0.8$ | $63.5 \pm 6.7$ | $68.3 \pm 2.1$ | $51.6 \pm 2.8$ | $45.2 \pm 4.4$ | $69.3 \pm 4.3$ | $60.0 \pm 5.5$ | $56.6 \pm 8.3$ |
| GNN-CL | $74.5 \pm 2.1$ | $59.5 \pm 2.4$ | $51.9 \pm 2.8$ | $68.2 \pm 2.0$ | $51.5 \pm 2.5$ | $42.6 \pm 3.2$ | $68.8 \pm 2.6$ | $59.4 \pm 3.4$ | $53.4 \pm 3.8$ |
| GraphENS | $78.7 \pm 0.6$ | $65.8 \pm 0.8$ | $63.1 \pm 1.0$ | $69.7 \pm 0.7$ | $53.5 \pm 0.9$ | $49.0 \pm 1.2$ | $72.7 \pm 3.0$ | $64.4 \pm 3.8$ | $61.6 \pm 4.5$ |
| LTE4G | $75.8 \pm 1.8$ | $61.5 \pm 2.6$ | $58.4 \pm 3.9$ | $66.8 \pm 1.5$ | $49.6 \pm 2.0$ | $45.6 \pm 3.1$ | $68.7 \pm 2.0$ | $59.3 \pm 1.6$ | $55.9 \pm 2.1$ |
| GraphSHA | $76.8 \pm 1.5$ | $62.9 \pm 2.1$ | $56.6 \pm 1.6$ | $65.7 \pm 1.2$ | $48.0 \pm 1.5$ | $39.9 \pm 1.6$ | $69.8 \pm 1.7$ | $60.7 \pm 2.2$ | $58.2 \pm 3.0$ |
| ReVar | $77.7 \pm 1.4$ | $64.3 \pm 2.1$ | $59.5 \pm 3.1$ | $69.4 \pm 0.3$ | $53.2 \pm 0.4$ | $46.6 \pm 0.5$ | $79.1 \pm 0.8$ | $72.5 \pm 1.0$ | $72.1 \pm 0.9$ |
| CBGG w/o $\mathcal{L}_{SC}$ (LTE4G) | $81.9 \pm 1.1$ | $70.5 \pm 1.7$ | $70.6 \pm 1.3$ | $73.5 \pm 2.6$ | $58.9 \pm 2.3$ | $57.7 \pm 1.8$ | $78.8 \pm 1.3$ | $72.1 \pm 1.7$ | $71.5 \pm 1.8$ |
| CBGG w/o $\mathcal{L}_{SC}$ (GraphENS) | $82.8 \pm 0.2$ | $71.9 \pm 0.3$ | $70.2 \pm 0.3$ | $72.9 \pm 1.3$ | $57.9 \pm 1.8$ | $53.6 \pm 0.5$ | $80.1 \pm 1.1$ | $73.8 \pm 1.5$ | $73.2 \pm 1.6$ |
| CBGG w/o $\mathcal{L}_{SC}$ (GraphSHA) | $82.6 \pm 0.3$ | $71.7 \pm 0.4$ | $70.4 \pm 0.3$ | $73.3 \pm 0.3$ | $58.6 \pm 0.4$ | $54.1 \pm 0.4$ | $79.8 \pm 0.7$ | $73.5 \pm 1.2$ | $73.1 \pm 1.2$ |
| CBGG (LTE4G) | $82.6 \pm 1.3$ | $71.6 \pm 2.0$ | $\underline{71.2 \pm 2.1}$ | $\mathbf{77.1 \pm 0.7}$ | $\underline{64.1 \pm 1.0}$ | $\mathbf{62.6 \pm 1.2}$ | $79.2 \pm 2.5$ | $72.7 \pm 3.3$ | $72.1 \pm 3.4$ |
| CBGG (GraphENS) | $\underline{83.2 \pm 0.2}$ | $\underline{72.5 \pm 0.3}$ | $70.7 \pm 0.3$ | $75.5 \pm 1.0$ | $61.8 \pm 1.4$ | $59.2 \pm 0.7$ | $\underline{81.0 \pm 0.6}$ | $\underline{75.0 \pm 0.8}$ | $\underline{74.7 \pm 0.8}$ |
| CBGG (GraphSHA) | $\mathbf{83.3 \pm 0.2}$ | $\mathbf{72.7 \pm 0.3}$ | $\mathbf{71.3 \pm 0.2}$ | $\underline{76.6 \pm 0.4}$ | $\mathbf{64.2 \pm 0.5}$ | $\underline{61.5 \pm 0.5}$ | $\mathbf{82.2 \pm 0.8}$ | $\mathbf{76.6 \pm 1.0}$ | $\mathbf{76.3 \pm 1.0}$ |

After all denoising steps, we obtain the final synthetic graph:

$$\widetilde{\mathcal{G}}_k := \widetilde{\mathcal{G}}_k^{(0)} = (\widetilde{\mathbf{X}}_k^{(0)}, \widetilde{\mathbf{A}}_k^{(0)}), \tag{11}$$

which is fully labeled by $\mathbf{Y}_k$ and structurally aligned with the original graph distribution.

**Training Node Classifier with Synthetic Graphs.** The final stage is to train the classifier $f_{\theta^*}$ with the class-balanced graphs $\{\widetilde{\mathcal{G}}_k\}_{k=1}^K$, along with the original graph $\mathcal{G}$. Since each synthetic graph contains approximately $N/C$ nodes per class due to uniform label sampling, the number of training nodes for both the most and least frequent classes increases by roughly $K \cdot N/C$. This augmentation makes the effective number of training nodes nearly identical across classes, thereby achieving class balance.

We train $f_{\theta^*}$ using the *Original-Synthetic Balanced Loss (OSBL)*, which combines supervision from both real and synthetic nodes:

$$\mathcal{L}_{\text{OSBL}} = \frac{1}{|\mathcal{V}_{\text{train}}|} \cdot \mathcal{L}_{\mathcal{G}} + \lambda \cdot \frac{1}{K} \sum_{k=1}^K \frac{1}{|\mathcal{V}^{\widehat{\mathcal{G}}_k}|} \cdot \mathcal{L}_{\widehat{\mathcal{G}}_k}, \tag{12}$$

where $\mathcal{L}_{\mathcal{G}}$ denotes the backbone model-specific loss for $f_{\theta^*}$, computed over the labeled nodes in $\mathcal{G}$, while $\mathcal{L}_{\widehat{\mathcal{G}}_k}$ denotes the same loss computed over the synthetic nodes in $\widehat{\mathcal{G}}_k$. $\lambda$ is a hyperparameter that controls the influence of the synthetic graphs in the overall OSBL. Trained under class-balanced conditions, the improved classifier $f_{\theta^*}$ produces the final output of CBGG: label predictions for the unlabeled nodes in $\mathcal{G}$.

### 3.5 THEORETICAL ANALYSIS

To understand how CBGG improves generalization in imbalanced node classification, we analyze its class-wise and overall generalization bounds. Let $f$ denote a node classifier. For each class $k \in \{1, \dots, C\}$, we define the class-conditional true risk as:

$$R_k(f) := \Pr(f(x) \neq k \mid y = k), \tag{13}$$

where $x \sim \mathcal{D}_k$, the true data distribution for class $k$.

Let $\tilde{\mathcal{D}}_k$ denote the distribution of synthetic nodes generated for class $k$ by the diffusion-based graph generator. We define the synthetic noise rate as:

$$\varepsilon_k^{\text{syn}} := \Pr_{x \sim \tilde{\mathcal{D}}_k} (f(x) \neq k). \tag{14}$$

CBGG explicitly minimizes the synthetic noise rate $\varepsilon_k^{\text{syn}}$ by training the graph generator using a supervised classification loss.

When training with both real and synthetic data, the generalization gap for class $k$ can be decomposed as:

$$\begin{aligned}
|R_k - \hat{R}_k| &\leq |R_k - \tilde{R}_k| + |\tilde{R}_k - \hat{R}_k| \\
&\leq \varepsilon_k^{\mathrm{syn}} + |\tilde{R}_k - \hat{R}_k| \\
&\leq \varepsilon_k^{\mathrm{syn}} + \sqrt{\frac{\log(2C/\delta)}{2\left(n_k + (1 - \varepsilon_k^{\mathrm{syn}})n_k^{\mathrm{syn}}\right)}}
\end{aligned} \tag{15}$$

(By Hoeffding's inequality).

Here, $\hat{R}_k$ is the empirical risk over synthetic samples and $\tilde{R}_k$ is the empirical risk over training data.

By using a class-balanced sampling, CBGG can substantially increase $n_k^{\mathrm{syn}}$ for all $k$. Therefore, the second term vanishes due to concentration, leaving the synthetic noise rate $\varepsilon_k^{\mathrm{syn}}$ as the dominant upper bound.

Consequently, the overall generalization gap of CBGG is bounded by the weighted average of synthetic noise rates across all classes:

$$|R - \hat{R}| \leq \sum_{k=1}^{C} p_k \varepsilon_k^{syn}, \tag{16}$$

where $p_k = \Pr(y = k)$ is the class prior.

This analysis shows that CBGG can reduce the generalization gap of $f$ by drastically increasing $n_k^{syn}$ through graph-level generation and by minimizing $\varepsilon_k^{syn}$ via the supervised contrastive loss in Eq.(8).

## 4 EXPERIMENTS

### 4.1 EXPERIMENTAL SETUP

**Datasets and Baselines.** We evaluate our proposed method on five widely used graph-structured datasets: Cora (McCallum et al., 2000), Citeseer (Giles et al., 1998), Pubmed (Sen et al., 2008), Amazon-Computers, and Amazon-Photo (Shchur et al., 2018). We compare our CBGG with various imbalance node classification methods, including three classic methods and six GNN-based methods. For classic methods, we select Over-sampling, Re-weight Yuan & Ma (2012), and SMOTE Chawla et al. (2002). For GNN-based methods, we choose GraphSMOTE Zhao et al. (2021), GNN-CL Li et al. (2024b), GraphENS Park et al. (2021a), LTE4G Yun et al. (2022), GraphSHA Li et al. (2023), and ReVar Yan et al. (2023). For fair comparisons, we set the hyperparameters of baselines according to specifications in the papers and official codes. For CBGG, we consistently set $\alpha$ and $\lambda$ to 0.1. Experimental details regarding datasets and baselines are provided in Appendix A.

**Evaluation Metrics.** In all experiments, we assess the performance of our CBGG and baselines with three metrics that are widely used and well-suited for class-imbalance classification: Geometric Mean (G-Means), Balanced Accuracy (bACC), and Macro-F1. Results are averaged over five independent runs with different random seeds, and we report the mean ± standard deviation.

### 4.2 EXPERIMENT RESULTS

**CBGG outperforms baselines by a large margin in step class imbalance settings.** We begin by evaluating the effectiveness of CBGG under step-class imbalance, where the imbalance ratio is standardized to $\rho = 0.05$ by assigning 20 training nodes to each head class and a single node to each tail class across all datasets. We instantiate CBGG using three different classifiers (LTE4G, GraphENS, and GraphSHA) and benchmark their performance against state-of-the-art methods to assess robustness across architectures. We standardize the imbalance level $\rho = 0.05$ across datasets by assigning 20 training nodes to each head class and only 1 to each tail class. As reported in Table 1 and Table 2, CBGG consistently outperforms baselines across all cases. On average, it yields relative improvements of 5.6% in G-Means and 8.3% in balanced accuracy over the strongest baseline, demonstrating the effectiveness of graph-level generation in imbalanced node classification.

Table 2: Step-class imbalanced node classification results on two Amazon networks. **Bold** marks the best scores, while underlining denotes the second-best.

| Method | Computers (Imb. class num = 5) | | Photo (Imb. class num = 3) | |
|---|---|---|---|---|
| **Imb. Ratio ($\rho = 0.05$)** | GMeans | bAcc | GMeans | bAcc |
| OverSampling | 68.1 ± 3.9 | 48.7 ± 5.8 | 85.5 ± 2.5 | 75.7 ± 4.0 |
| Reweight | 77.7 ± 1.7 | 62.9 ± 2.5 | 88.6 ± 3.3 | 80.7 ± 3.2 |
| SMOTE | 76.8 ± 2.1 | 61.6 ± 3.1 | 87.1 ± 2.5 | 78.4 ± 4.1 |
| GraphSMOTE | 77.9 ± 4.5 | 63.4 ± 6.8 | 85.6 ± 3.6 | 75.9 ± 5.7 |
| GNN-CL | 77.6 ± 3.1 | 64.5 ± 3.4 | 86.9 ± 2.8 | 78.0 ± 3.1 |
| GraphENS | 77.8 ± 0.8 | 63.1 ± 1.1 | 91.3 ± 0.3 | 84.8 ± 0.5 |
| LTE4G | 79.3 ± 4.6 | 64.4 ± 3.9 | 86.5 ± 5.1 | 77.4 ± 8.2 |
| GraphSHA | 78.4 ± 0.4 | 63.9 ± 0.7 | 86.9 ± 1.5 | 78.0 ± 2.4 |
| ReVar | 81.3 ± 1.6 | 75.1 ± 2.1 | 89.9 ± 0.7 | 82.9 ± 1.2 |
| CBGG w/o $\mathcal{L}_{SC}$ (LTE4G) | 83.1 ± 1.2 | 74.2 ± 2.2 | 88.9 ± 1.0 | 81.2 ± 1.6 |
| CBGG w/o $\mathcal{L}_{SC}$ (GraphENS) | 84.5 ± 0.5 | 73.6 ± 2.1 | 92.1 ± 0.2 | 86.4 ± 0.4 |
| CBGG w/o $\mathcal{L}_{SC}$ (GraphSHA) | 83.3 ± 1.0 | 71.7 ± 1.6 | 90.5 ± 0.5 | 83.9 ± 0.8 |
| CBGG (LTE4G) | 83.6 ± 1.3 | 74.6 ± 2.0 | 92.2 ± 2.1 | 86.7 ± 3.5 |
| CBGG (GraphENS) | 85.2 ± 1.2 | 75.5 ± 0.3 | 92.8 ± 0.3 | 87.7 ± 0.4 |
| CBGG (GraphSHA) | 84.9 ± 1.4 | 74.2 ± 2.2 | 91.3 ± 0.7 | 85.2 ± 1.1 |

Table 3: Long-tailed node classification results on two citation networks. **Bold** marks the best scores, while underlining denotes the second-best.

| Method | Cora-LT | | CiteSeer-LT | |
|---|---|---|---|---|
| **Imb. Ratio ($\rho = 0.01$)** | GMeans | bAcc | GMeans | bAcc |
| OverSampling | 77.2 ± 1.7 | 63.5 ± 2.6 | 65.6 ± 1.5 | 48.0 ± 2.0 |
| Reweight | 81.2 ± 0.9 | 69.4 ± 1.3 | 70.6 ± 0.8 | 54.8 ± 1.1 |
| SMOTE | 76.3 ± 1.5 | 62.2 ± 2.1 | 64.4 ± 2.9 | 46.5 ± 3.7 |
| GraphSMOTE | 80.0 ± 0.9 | 67.6 ± 1.4 | 65.5 ± 0.9 | 47.9 ± 1.2 |
| GNN-CL | 80.2 ± 0.7 | 66.9 ± 1.0 | 71.2 ± 0.4 | 55.7 ± 0.6 |
| GraphENS | 83.7 ± 0.3 | 73.5 ± 0.5 | 73.8 ± 0.5 | 59.3 ± 0.2 |
| LTE4G | 80.2 ± 1.4 | 67.9 ± 2.2 | 71.5 ± 1.4 | 56.1 ± 2.0 |
| GraphSHA | 84.5 ± 1.0 | 74.6 ± 1.5 | 73.4 ± 0.6 | 58.7 ± 0.9 |
| ReVar | 81.0 ± 0.4 | 69.1 ± 0.6 | 73.7 ± 0.4 | 59.2 ± 0.5 |
| CBGG w/o $\mathcal{L}_{SC}$ (LTE4G) | 82.7 ± 0.8 | 71.8 ± 1.3 | 75.2 ± 0.6 | 61.3 ± 0.9 |
| CBGG w/o $\mathcal{L}_{SC}$ (GraphENS) | 85.5 ± 0.3 | 76.2 ± 0.5 | 75.1 ± 0.1 | 61.3 ± 0.2 |
| CBGG w/o $\mathcal{L}_{SC}$ (GraphSHA) | 85.5 ± 0.3 | 76.2 ± 0.5 | 75.0 ± 0.3 | 61.0 ± 0.4 |
| CBGG (LTE4G) | 83.1 ± 0.2 | 72.4 ± 0.3 | 75.3 ± 0.2 | 61.4 ± 0.3 |
| CBGG (GraphENS) | 87.1 ± 0.2 | 78.7 ± 0.3 | 75.2 ± 0.1 | 61.3 ± 0.2 |
| CBGG (GraphSHA) | 85.9 ± 0.3 | 76.9 ± 0.4 | 75.4 ± 0.2 | 61.6 ± 0.3 |

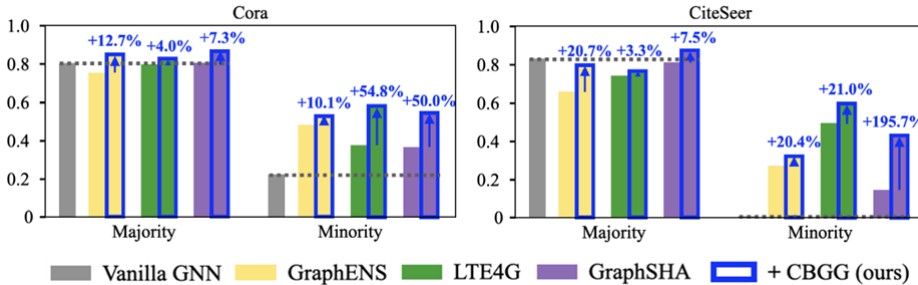

Figure 3: Recall of majority and minority classes on Cora and CiteSeer.

**CBGG consistently outperforms baselines in long-tailed imbalance settings.** We further evaluate the performance of CBGG under long-tailed class distributions. For each dataset, we sort the classes in descending order of frequency, then iteratively remove low-degree nodes from the minority classes until the desired imbalance ratio $\rho = 0.01$ is reached. Here, $\rho$ denotes the ratio between the most frequent class and the least frequent class. As shown in Table 3, CBGG consistently outperforms all competing methods under long-tailed class distributions, further demonstrating its robustness to severe class imbalance.

**CBGG effectively improves the performance of imbalanced node classifiers.** CBGG aims to improve the performance of imbalanced node classifiers by leveraging class-balanced synthetic graphs generated by a graph diffusion process. To assess its effectiveness, we instantiate CBGG with three different downstream classifiers (LTE4G, GraphENS, and GraphSHA) under both step-class and long-tailed imbalance settings. As shown in Tables 1, 2, and 3, CBGG consistently improves performance across all classifiers and imbalance conditions.

$\mathcal{L}_{SC}$ **improves the performance of CBGG.** Unlike prior diffusion-based graph generators, our graph generator $g_\phi$ leans to minimize $\mathcal{L}_{SC}$ to support node classification. By tightly clustering the representative points of the same class while pushing different classes farther apart, it reduces boundary instability arising from label noise. To isolate the effect of $\mathcal{L}_{SC}$, we set $\alpha = 0$ and denote this ablation as $\text{CBGG}_{w/o\mathcal{L}_{SC}}$. As reported in Tables 1, 2, and 3, this variant consistently underperforms compared to CBGG. These results supports that including $\mathcal{L}_{SC}$ effectively reduces the synthetic error in Eq.(14) and thus yields an improvement in classification accuracy.

**CBGG mitigates the majority-minority performance tradeoff of imbalanced node classifiers.** Training on class-balanced synthetic graphs, CBGG increases the recall of majority classes without sacrificing the accuracy of minority classes. To verify this effect, we measure the recall of three imbalanced node classifiers (LTE4G, GraphENS, and GraphSHA) and their CBGG-augmented counterparts. As shown in Figure 3, CBGG consistently and substantially improves recall for majority classes across all classifiers. These results demonstrate that class-balanced synthetic graphs

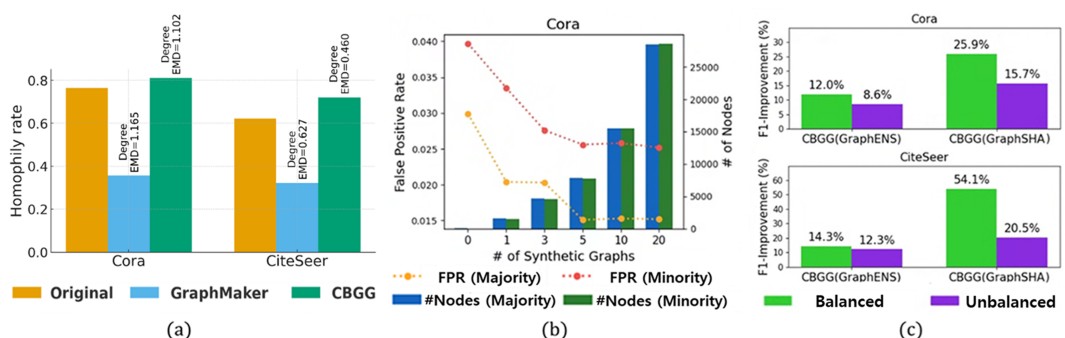

Figure 4: (a) Homophily rate and degree earth mover's distance of synthetic graphs, (b) false positive rate and the number of training nodes depending on the number of $\widehat{\mathcal{G}}_k$, and (c) relative G-Means improvement (%) over GraphENS and GraphSHA.

effectively mitigate class imbalance, enabling improved overall performance without compromising majority-class accuracy.

**Does the generator $g_\phi$ capture realistic graph properties?** Our CBGG introduces a new diffusion-based graph generator that conditions soft-label distributions of nodes. To investigate if CBGG reproduces structural properties of real-world graphs, we assess the fidelity of its generated graphs against those of GraphMaker (Li et al., 2024a). Our evaluation uses two metrics: (i) the homophily rate, which measures assortativity by class, and (ii) the Degree Earth Mover's Distance (Degree EMD), which quantifies the dissimilarity between the node degree distributions of original and synthetic graphs. Figure 4(a) shows that CBGG generates synthetic graphs with a homophily rate nearly identical to the original graph, whereas GraphMaker fails to capture this fundamental property. Furthermore, CBGG consistently achieves a lower Degree EMD than GraphMaker on both Cora and CiteSeer, indicating a more faithful reproduction of the node degree distribution.

**Does the number of synthetic graphs matter?** To further examine the impact of class-balanced synthetic graphs, we vary the number of generated graphs $K \in \{0, 1, 3, 5, 10, 20\}$ and measure the false positive rate (FPR) of CBGG. Figure 4(b) shows that adding more synthetic graphs consistently decreases the FPR of both majority and minority nodes. Notably, the effect saturates when $K \geq 5$, suggesting that a moderate number of synthetic graphs is sufficient.

**Class-balanced synthetic graphs outperform unbalanced counterparts.** To assess the impact of class-balanced sampling, we create a variant of CBGG where node labels are drawn from the empirical distribution of $\widehat{\mathbf{Y}}$, instead of the uniform prior in Eq. (9). We then train the classifiers $f_{\theta^*}$ (GraphENS and GraphSHA) on both balanced and unbalanced synthetic graphs, and compare their F1-score improvement over initial classifiers $f_\theta$. As shown in Figure 4(c), balanced sampling consistently leads to gains across both classifiers. These results are consistent with our theoretical analysis in Eq.(15): balanced sampling increases the minimum class-wise sample count $n_k$, which tightens the generalization bound across all classes.

Additional experiments are provided in the Appendix, including hyperparameter analysis (in Appendix C), loss analysis (in Appendix D), time complexity analysis (in Appendix E), and robustness to noise (in Appendix F).

## 5 CONCLUSION

In this work, we propose Class Balancing via Graph Generation (CBGG), a novel framework for imbalanced node classification that alleviates the majority–minority trade-off by generating class-balanced synthetic graphs. We theoretically and empirically show that CBGG narrows the generalization gap by (i) increasing the number of training samples through graph-level genera-tion and (ii) reducing synthetic noise via a supervised contrastive loss. Extensive experiments on seven benchmarks confirm that CBGG substantially improves the performance of state-of-the-art methods.We expect that our graph-level generation–based approach will serve as a new direction for addressing imbalanced node classification. While our study focuses on homogeneous graphs, extending this approach to heterogeneous graphs is a promising future direction.

## REPRODUCIBILITY STATEMENT

We have made careful efforts to ensure the reproducibility of our work. Specifically, Sec. 4.1 describes the experimental setup, including datasets, baselines, and evaluation metrics. Detailed descriptions of the experimental settings, including hardware specifications and software environments, are provided in Appendix A. Dataset information and download sources are given in Appendix A.2, while baseline implementations and hyperparameter choices are detailed in Appendix A.3. Finally, definitions of all evaluation metrics are summarized in Appendix A.3. The source code is provided in the supplementary material and will be made publicly available upon publication.

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

# A    EXPERIMENTAL DETAILS

## A.1    SETTINGS

We conduct experiments on four NVIDIA GeForce RTX 3090 Ti GPUs and an Intel Core i9-12900k CPU. We use CUDA 11.8, Python 3.9.21, PyTorch 2.0.0+cu118 (Paszke et al., 2019), and PyTorch Geometric 2.3.1 (Fey & Lenssen, 2019).

## A.2    DATASETS

We conduct experiments on five benchmark graph-structured datasets: Cora (McCallum et al., 2000), CiteSeer (Giles et al., 1998), PubMed (Sen et al., 2008), Amazon-Computers (Shchur et al., 2018), and Amazon-Photo (Shchur et al., 2018). The three citation datasets (Cora, Cite-seer, PubMed) are obtained from the PyTorch Geometric repository [1]. The two Amazon graphs are also downloaded from the corresponding loader in the same repository [2]. These publicly available datasets and the repository have no public declaration of license.

Table 4: Dataset statistics.

| Dataset | # of Nodes | # of Edges | # of Features | # of Classes | # of Imbalanced Classes |
|---|---|---|---|---|---|
| Cora | 2,708 | 5,429 | 1,433 | 7 | 3 |
| CiteSeer | 3,327 | 4,732 | 3,703 | 6 | 3 |
| PubMed | 19,717 | 44,338 | 500 | 3 | 1 |
| Amazon Computers | 13,381 | 245,778 | 767 | 10 | 5 |
| Amazon Photo | 7,487 | 119,043 | 745 | 8 | 3 |

## A.3    BASELINES

For each baseline, we utilize the following publicly available codes. We select the hyperparameter setting for each baseline in the released code or its respective paper. For GraphENS, we use $Beta(2,2)$ distribution to sample its mixing ratio. For GraphSHA, we choose the graph diffusion as the Personalized PageRank (PPR) with a teleport rate $\alpha = 0.05$ and then select 128 highest mass per column. For LTE4G, following the original protocol, we designate 7 head / 3 tail classes on Cora and 5 head / 5 tail classes on Citeseer in long-tailed settings.

Table 5: URL links for baselines.

| Baseline | URL Link |
|---|---|
| Oversampling | `https://github.com/SukwonYun/LTE4G/blob/main/models/baseline/oversampling.py` |
| Reweighting | `https://github.com/SukwonYun/LTE4G/blob/main/models/baseline/reweight.py` |
| SMOTE | `https://github.com/SukwonYun/LTE4G/blob/main/models/baseline/smote.py` |
| GraphSMOTE | `https://github.com/SukwonYun/LTE4G/blob/main/models/baseline/graphsmote_T.py` |
| GNN-CL | `https://github.com/seanlxh/GNN-CL` |
| GraphENS | `https://github.com/JoonHyung-Park/GraphENS` |
| LTE4G | `https://github.com/SukwonYun/LTE4G` |
| GraphSHA | `https://github.com/wenzhilics/GraphSHA` |
| ReVar | `https://github.com/yanliang3612/ReVar` |

## A.4    EVALUATION METRICS

We use three different metrics: Geometric Means (G-Means), Balanced Accuracy (bAcc), and Macro-F1. Here, higher G-Means, bAcc, and Macro-F1 indicate better performance.

- **G-Means.** The geometric mean of the true-positive rate (sensitivity) and true-negative rate (specificity). It rewards models that perform well on both the positive and negative classes,

---

[1] `https://github.com/pyg-team/pytorch_geometric/blob/master/torch_geometric/datasets/planetoid.py`

[2] `https://github.com/pyg-team/pytorch_geometric/blob/master/torch_geometric/datasets/amazon.py`

Table 6: Imbalanced node classification results on three citation networks. For each method, we report the mean ± standard error over five runs for geometric mean (G-Means %), balanced accuracy (bAcc %), and macro-F1 score. **Bold** marks the best scores, while underlining denotes the second-best.

| Method | Cora (Imb. class num = 3) | | | Citeseer (Imb. class num = 3) | | | Pubmed (Imb. class num = 1) | | |
|---|---|---|---|---|---|---|---|---|---|
| Imb. Ratio ($\rho = 0.1$) | GMeans | bAcc | Macro-F1 | GMeans | bAcc | Macro-F1 | GMeans | bAcc | Macro-F1 |
| OverSampling | 74.4 ± 2.9 | 59.4 ± 4.2 | 55.5 ± 6.6 | 62.2 ± 2.1 | 43.6 ± 2.7 | 34.1 ± 3.1 | 68.5 ± 0.9 | 59.0 ± 1.1 | 54.0 ± 4.5 |
| Reweight | 81.3 ± 3.4 | 70.2 ± 5.1 | 70.3 ± 5.9 | 67.1 ± 0.8 | 50.0 ± 1.1 | 43.3 ± 1.4 | 75.0 ± 0.4 | 67.3 ± 0.5 | 66.1 ± 0.3 |
| SMOTE | 74.0 ± 1.5 | 58.8 ± 2.2 | 55.4 ± 2.6 | 63.9 ± 1.0 | 45.8 ± 1.3 | 37.9 ± 1.6 | 71.0 ± 2.7 | 62.2 ± 3.5 | 58.0 ± 6.7 |
| GraphSMOTE | 82.1 ± 0.6 | 71.0 ± 5.9 | 69.4 ± 7.9 | 67.0 ± 1.7 | 49.9 ± 2.2 | 44.2 ± 3.2 | 74.7 ± 2.1 | 66.9 ± 2.7 | 65.2 ± 3.3 |
| GNN-CL | 82.5 ± 1.1 | 72.0 ± 1.7 | 65.3 ± 1.5 | 67.8 ± 0.4 | 50.8 ± 0.6 | 50.1 ± 0.1 | 80.8 ± 1.6 | 74.7 ± 2.0 | 74.1 ± 2.7 |
| GraphENS | 79.3 ± 0.8 | 69.6 ± 1.3 | 68.6 ± 1.6 | 71.7 ± 1.1 | 56.4 ± 1.6 | 49.2 ± 2.9 | 80.5 ± 1.6 | 74.3 ± 2.0 | 74.3 ± 2.0 |
| LTE4G | 83.6 ± 3.1 | 73.2 ± 3.9 | 72.1 ± 5.8 | 70.2 ± 3.3 | 54.0 ± 4.1 | 51.8 ± 4.0 | 73.7 ± 1.4 | 65.7 ± 1.7 | 65.2 ± 1.3 |
| GraphSHA | 85.3 ± 0.3 | 75.8 ± 0.4 | 75.7 ± 0.4 | 72.3 ± 1.7 | 56.6 ± 2.4 | 56.3 ± 2.1 | 81.9 ± 0.2 | 76.1 ± 0.3 | 75.8 ± 0.4 |
| ReVar | 83.7 ± 1.1 | 73.4 ± 1.7 | 72.7 ± 1.9 | 72.4 ± 1.6 | 57.3 ± 2.3 | 53.8 ± 1.7 | 81.9 ± 0.9 | 76.1 ± 1.1 | 76.1 ± 1.2 |
| CBGG (GraphENS) | 85.7 ± 0.5 | 76.1 ± 0.7 | 76.0 ± 0.8 | **78.1 ± 0.4** | **65.5 ± 0.6** | 62.1 ± 0.5 | 82.7 ± 1.8 | 77.2 ± 2.3 | 77.0 ± 2.2 |
| CBGG (GraphSHA) | **86.7 ± 0.1** | **78.1 ± 0.2** | **78.2 ± 0.2** | 77.5 ± 0.4 | 64.6 ± 0.5 | **62.7 ± 0.6** | **84.8 ± 0.4** | **79.9 ± 0.6** | **79.6 ± 0.5** |

even under severe class imbalance:

$$\text{G-Means} = \sqrt{\frac{\text{TP}}{\text{TP} + \text{FN}} \times \frac{\text{TN}}{\text{TN} + \text{FP}}} \tag{17}$$

where TP is the number of true positives, TN is the number of true negatives, FP is the number of false positives, and FN is the number of false negatives.

- **bAcc.** The arithmetic mean of per-class recalls, so every class contributes equally regardless of its prevalence:

$$\text{bAcc} = \frac{1}{C} \sum_{c=1}^{C} \frac{\text{TP}_c}{\text{TP}_c + \text{FN}_c}, \tag{18}$$

where $\text{TP}_c$ is the number of true positives on the class $c$, and $\text{FN}_c$ is the number of false negatives on the class $c$.

- **Macro-F1.** Macro-F1 calculates the F1-score for each class and then takes their arithmetic mean, so that precision and recall are both considered for every class as:

$$\text{Macro-F1} = \frac{1}{C} \sum_{c=1}^{C} \frac{2\text{Precision}_c\text{Recall}_c}{\text{Precision}_c + \text{Recall}_c} \tag{19}$$

where $\text{Precision}_c = \frac{\text{TP}_c}{\text{TP}_c + \text{FP}_c}$, and $\text{Recall}_c = \frac{\text{TP}_c}{\text{TP}_c + \text{FN}_c}$.

## B    RESULTS ON THE ANOTHER IMBALANCED RATE ($\rho = 0.1$)

We further assess the effectiveness of CBGG under another step-class imbalance of $\rho = 0.1$. Following the setup in Section 4.2, we instantiate CBGG with two final classifiers $f_{\theta*}$ (GraphENS and GraphSHA), then compare them against state-of-the-art baselines. Following the practice of Park et al. (2021a); Yun et al. (2022), we equalise the imbalance level across all datasets: each *head* class is given 20 labelled nodes, whereas each *tail* class receives only two. All other hyperparameters and training settings remain unchanged from Section 4.2. Table 6 shows that CBGG continues to outperform every baseline on every dataset and metric, even under this milder imbalance. Moreover, both GraphENS and GraphSHA benefit from CBGG, exhibiting consistent improvements in all metrics. These results confirm that the gains delivered by our class-balancing diffusion module are not confined to the most extreme imbalance setting, but generalise to more moderate scenarios as well.

## C    HYPERPARAMETER ANALYSIS

CBGG employs two key hyperparameters: $\lambda$, which controls how strongly the synthetic graphs influence classifier training, and $\alpha$, which weights the supervised contrastive loss $\mathcal{L}_{\text{SC}}$ loss within

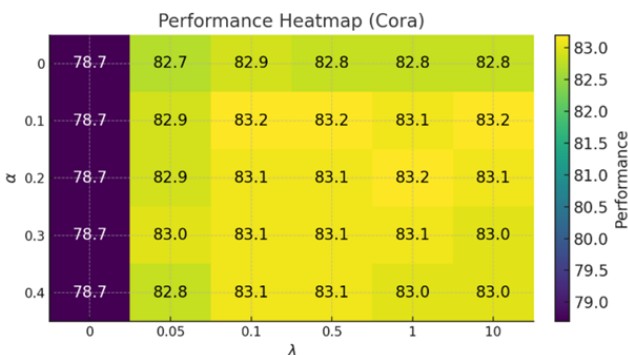

Figure 5: Hyperparameter sensitivity ($\alpha$ and $\lambda$) of CBGG.

the graph-generation objective. To assess its impact, we vary $\alpha \in \{0, 0.1, 0.2, 0.3, 0.4\}$ and $\lambda \in \{0, 0.05, 0.1, 0.5, 1, 10\}$ and report the resulting G-Means scores of CBGG in Fig. 5. Each setting is repeated five times, and the averages are plotted. When $\lambda=0$, CBGG invariably attains the lowest score, confirming that *including synthetic graphs is essential*. Performance also drops slightly at $\alpha=0$, because an overly small $\alpha$ cannot sufficiently mitigate class imbalance. The sweet spot lies in the mid-range $\alpha \in [0.1, 0.3]$ and $\lambda \in [0.05, 1]$, where CBGG consistently reaches its global maximum (~83.2). Since this optimal region is broad, hyperparameter tuning incurs little overhead and readily scales to larger datasets.

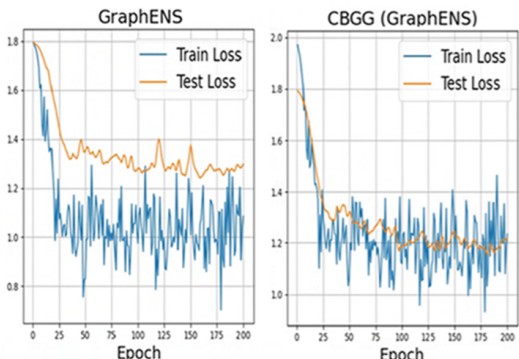

Figure 6: Train and test loss.

## D  LOSS ANALYSIS

To examine the effect of synthetic graph generation on generalization, we compare the training and test losses of vanilla GRAPHENS and CBGG that uses the same classifier, as shown in Figure 6. Across five random seeds (mean shown), CBGG consistently exhibits a much narrower training–test gap, implying a smaller generalization gap than the vanilla model. We attribute this improvement to the substantially larger number of nodes introduced by graph generation, which acts as an on-the-fly data-augmentation mechanism and regularizes the classifier.

## E  TIME COMPLEXITY ANALYSIS

We analyze the computational complexity of the proposed CBGG framework. The graph generator $g_\psi$ is implemented as a single-layer Graph Attention Network (GAT) (Veličković et al., 2018) with embedding dimension $H$. During generation, we employ minibatch sampling, where each denoising step processes $N_b$ nodes and $E_b$ edges, resulting in a per-step time complexity of $\mathcal{O}(N_b H^2 + E_b H)$. To generate $K$ synthetic graphs, the reverse diffusion process is repeated

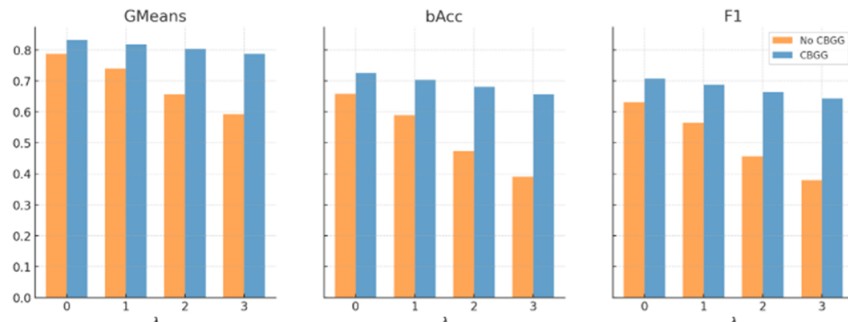

Figure 7: Performance variation by noise injection level in the initial classifier's soft label distribution

for $T$ steps per graph, leading to a total sampling cost of $\mathcal{O}(KT(N_bH^2 + E_bH))$. Additionally, training the final classifier $f_{\theta^*}$ on both the real and synthetic graphs incurs a complexity $\mathcal{O}((1+K)L(NH^2+EH))$ where $L$ denotes the number of GNN layers. Then, the overall complexity of CBGG becomes $\mathcal{O}\left(KT(N_bH^2 + E_bH) + (1 + K)L(NH^2 + EH)\right)$. Under typical settings such as $H$=256, $N$=$10^4$, $E$=$10^5$, $T$=30, $K$=5, and minibatch size $N_b$=$E_b$=500, the training cost of CBGG is approximately 7.5 times higher than that of a standard GNN. At inference time, however, CBGG requires only a single forward pass of the trained classifier on the original graph, incurring the same computational cost as a vanilla GNN while delivering substantial performance gains. Despite the additional training cost, CBGG offers a practical solution that delivers significant performance improvements without increasing inference cost.

## F ROBUSTNESS TO NOISE

In real-world scenarios, the initial classifier often produces noisy predictions. To evaluate whether CBGG is robust to such imperfect supervision, we inject Gaussian noise of varying magnitude ($\lambda \in 0, 1, 2, 3$) into the teacher outputs. Figure 7 presents the results on a GraphENS classifier and its improved classifier via CBGG. As the noise level increases, the vanilla GraphENS baseline experiences a sharp decline in G-Means, balanced accuracy, and Macro-F1. In contrast, the CBGG-augmented models remain substantially more stable, maintaining high performance even under the strongest perturbations. These findings indicate that CBGG effectively mitigates the impact of noisy teacher signals, thereby providing robustness against unreliable initial supervision.

