# OpenReview forum: "Why Sacrifice Majority Nodes?: Improving Imbalanced Node Classification via Class-Balanced Graph Generation"
_ICLR.cc/2026/Conference — ICLR 2026 Conference Withdrawn Submission_

### Official Review · Reviewer_bcCo · 2025-10-28

**Soundness:** 2
**Presentation:** 3
**Contribution:** 2
**Rating:** 4
**Confidence:** 4

**Summary:**

This paper focuses on the problem of class-imbalanced node classification and considers the trade-off between the majority and minority classes in existing oversampling methods.
To avoid sacrificing predictive power for the majority classes, this paper proposes Class-Balanced Graph Generation (CBGG). First, CBGG trains a novel diffusion-based graph generator, conditioned on the soft labels of an initial classifier, to synthesize  class-balanced graphs. Then, CBGG retrains the classifier based on the class-balanced graphs, thereby improving class-imbalanced node classification.
Extensive experimental results demonstrate the effectiveness of the proposed approach.

**Strengths:**

1. This paper is well-organized and easy-to-follow.
2. The experimental results show the proposed method can achieve state-ot-the-art performance.

**Weaknesses:**

1. The novelty of the proposed method seems limited. The diffusion-based graph generator follows the previous method GraphMaker, including in the forward process and reverse process. CBGG generates graphs conditioned on the soft labels, which is also discusses in GraphMarker, "To generate graph data with node labels, one naïve way is to simply generate node labels as extra node attributes" (See Section 2.4 'Conditional generation given node labels' in GraphMarker).  Could you show some difference of the proposed graph generator compared with that in GraphMaker?
2. As shown in Figure 1b, CGBB generates a large amount of training data, such as tens of thousands of training nodes, while other baselines that also use oversampling only have dozens of training nodes. Is this a fair comparison?
3. The diffusion-based graph generator is conditioned on soft labels of an initial classifier, however, is the soft labels are reliable for graph generation? In particular, the classifier may under-perform on minority nodes, could the unreliable soft labels impact the graph generation?
4. Due to the training cost of CGBB, the datasets used in the experiments are relatively small and cannot be well applied to real-world scenarios. In order to verify the practicality of CGBB, it should be extended to large datasets, such as the OGBN dataset.

**Questions:**

See Weaknesses.

---

### Official Review · Reviewer_zwKq · 2025-10-29

**Soundness:** 3
**Presentation:** 3
**Contribution:** 2
**Rating:** 2
**Confidence:** 3

**Summary:**

This paper studies the class imbalance problem on graph data. In specific, it studies the imbalance problem for node classification tasks. Its core idea can be summarized as

1. a generative model (a diffusion model in this paper) is trained on the given graph, conditioned on the node label matrix
2. adjust the node label matrix so the labels are balanced; feed the balanced node label matrix to the generative model to generate "label-balance" synthetic graphs
3. train a node classifier on both the original graph and the synthetic graphs.

**Strengths:**

S1. The overall method is easy to understand. The presentation of this paper is good.

S2. The core idea is intuitive and reasonable, which generates the balanced class distribution, which improves the classifier's performance.

S3. Looks like the proposed method gains good performance compared to baseline methods (Table 2)

**Weaknesses:**

W1. The novelty of this paper is not significant. Fundamentally, the key idea, as mentioned above, is to generate a balanced class distribution, which has been widely used in many literatures (on node classification problem) such as using mixup, SMOTE.

W2. It is concerning why the proposed method can work: the method applies a diffusion model trained on the given graph. However, for the node classification tasks, the input graph usually only includes 1 graph. In other words, the # of training sample for the diffusion model is only 1. In that case, the model will tend to over-memorize the given data sample.

W3. The contribution of this paper, to be frank, is limited. In detail

W3.1 a lot of the section 3.3 is established model design. The main uniqueness is that the diffusion model is conditioned on the "node label matrix", different from the typical setting which conditions on the "graph label".

W3.2 The socalled "Original-Synthetic Balanced Loss (OSBL)" is nothing new. It is very common to train the classifier on both the original nodes and synthetic nodes for the imbalanced classification problem.

W3.3 The theoretical analysis (section 3.5) is not a unique contribution of this paper, which applies to most "rebalancing" based method, e.g., SMOTE.

W4. I think the model architecture is not talked. I suggest to introduce it

**Questions:**

Q1. The paper mentioned that the conditioning is based on "concatenating soft-label label vectors". If I understand correctly, it concatenates the label vectors from all the nodes which would be pretty long. How the model architecture is designed to handle it?

Q2. Above Eq. 9, the paper mentioned that it first sampled class c_i and then"independently sampling soft label vectors" based on c_i. It is a bit vague how define the label of the "soft label vectors"? Is it by its largest logit?

---

### Official Review · Reviewer_gEJA · 2025-11-01

**Soundness:** 3
**Presentation:** 3
**Contribution:** 3
**Rating:** 6
**Confidence:** 4

**Summary:**

This paper introduces CBGG, a class-balanced graph generation framework that tackles the performance trade-off in imbalanced node classification where boosting minority recall typically compromises majority recall. Through a conditional diffusion model, CBGG produces high-quality synthetic graphs with balanced class distributions, providing diverse training samples for all classes. Comprehensive evaluations across seven benchmarks show CBGG substantially outperforms state-of-the-art methods while effectively alleviating the trade-off, simultaneously enhancing performance for both majority and minority classes. This work establishes graph-level generation as a new paradigm for class imbalance, supported by theoretical generalization bounds, offering significant implications and a solid benchmark for future studies.

**Strengths:**

1. This paper systematically uncovers the long-overlooked "performance trade-off" in imbalanced graph learning, shifting the research focus from partially optimizing minority classes to holistically enhancing all categories.
2. Paper pioneers a graph-level generation paradigm using conditional diffusion models to create class-balanced synthetic graphs, fundamentally redefining the solution landscape beyond conventional node-level oversampling.
3. The method innovatively integrates soft-label conditioning with supervised contrastive learning, achieving deep synergy between the graph generation process and node classification objectives.
4. The study establishes a comprehensive framework of argumentation integrating methodology, theoretical analysis, and experimental validation, demonstrating performance superiority across benchmark datasets while providing rigorous explanations through generalization theory.

**Weaknesses:**

1. The survey of related work is somewhat inadequate, particularly lacking further elaboration on whether existing studies have attempted to address the inherent minority-majority trade-off problem.
2. It is suggested to roughly categorize the experimental results by type (e.g., comparative experiments, analytical experiments) to enhance the structural clarity. Although experiments have been conducted under various scenarios, there remains a lack of additional analytical experiments to validate the model's superiority.

**Questions:**

1. The paper mentions that existing methods have not sufficiently explored the inherent minority–majority trade-off. Are there any methods that have attempted to address this issue? If so, please summarize such methods and compare them with the method proposed in this paper to further argue the motivation of this chapter.
2. In line 192, there is a lack of explanation for the function γ_Z(t).
3. In the paragraph corresponding to line 450 in Section 4.2, two metrics are used to evaluate whether the generator can capture real graph properties. These two metrics are mainly used to measure the differences in node attributes, but there is a lack of metrics to measure the semantic differences in edges.
4. The core innovation of this paper is graph-level generation. Can experiments be designed to isolate the effect of generation itself? For example, comparing it with a simple method of “sampling nodes from the original graph to form a new graph” to prove that the high-quality data generated by the diffusion model is the key to performance improvement, rather than simple data augmentation.

---

### Official Review · Reviewer_iUpF · 2025-11-02

**Soundness:** 2
**Presentation:** 2
**Contribution:** 2
**Rating:** 4
**Confidence:** 3

**Summary:**

This paper addresses the critical issue of class imbalance in graph-structured data for node classification. The authors identify a key limitation in existing methods: while they improve minority-class recall, they often do so at the expense of majority-class performance. To mitigate this trade-off, the proposed framework, Class Balancing Graph Generation (CBGG), leverages diffusion-based graph generation to create class-balanced synthetic graphs. By training classifiers on these graphs, CBGG aims to tighten generalization bounds across all classes. The authors support their claims with theoretical analysis and extensive experiments on five benchmark datasets, demonstrating state-of-the-art performance.

**Strengths:**

1. The paper highlights an underexplored but important trade-off in imbalanced node classification—improving minority recall at the cost of majority recall—which motivates the proposed method.

2. CBGG introduces a graph-level generation paradigm for class imbalance, combining diffusion-based generation with supervised contrastive learning to produce high-quality synthetic graphs.

3. The authors provide a theoretical analysis of generalization bounds and empirically validate CBGG on seven benchmarks, showing consistent improvements over existing methods.

**Weaknesses:**

1. The experimental evaluation, while extensive, does not include large-scale real-world graphs (e.g., with millions of nodes), raising questions about scalability and applicability to truly massive datasets.

2. The use of diffusion-based graph generation and multiple training stages (initial classifier, generator, and final classifier) may introduce significant computational overhead, which is not thoroughly discussed or evaluated.

**Questions:**

1. How does CBGG ensure the quality and diversity of synthetic graphs, especially when the initial classifier is trained on imbalanced data?

2. Could the performance gains be attributed solely to the increased quantity of synthetic data, rather than the graph generation process itself?

---

### Note · Authors · 2025-11-16

I have read and agree with the venue's withdrawal policy on behalf of myself and my co-authors.